# A Scoping Review of Selected Studies on Predictor Variables Associated with the Malaria Status among Children under Five Years in Sub-Saharan Africa

**DOI:** 10.3390/ijerph18042119

**Published:** 2021-02-22

**Authors:** Phillips Edomwonyi Obasohan, Stephen J. Walters, Richard Jacques, Khaled Khatab

**Affiliations:** 1School of Health and Related Research (ScHARR), University of Sheffield, Sheffield S1 4AD, UK; s.j.walters@sheffield.ac.uk (S.J.W.); r.jacques@sheffield.ac.uk (R.J.); 2Department of Liberal Studies, College of Business and Administrative Studies, Niger State Polytechnic, Bida Campus, Bida 912231, Nigeria; 3Faculty of Health and Wellbeing, Sheffield Hallam University, Sheffield S10 2BP, UK; k.khatab@shu.ac.uk

**Keywords:** malaria, fever, *Plasmodium falciparum*, *Falciparum vivax*, under-five, determinants, risk factors, review

## Abstract

Background/Purpose: In recent times, Sub-Saharan Africa (SSA) had been rated by the World Health Organization (WHO) as the most malaria-endemic region in the world. Evidence synthesis of the factors associated with malaria among children aged under-five in SSA is urgently needed. This would help to inform decisions that policymakers and executors in the region need to make for the effective distribution of scarce palliative resources to curb the spread of the illness. This scoping review is aimed at identifying studies that have used multivariate classical regression analysis to determine the predictors associated with malaria among children under five years old in SSA. Methods/Design: The search terms followed population, intervention, comparator, outcome, timing, setting (PICOTS), and were used in searching through the following databases: PubMed, MEDLINE, Web of Science, Cumulative Index of Nursing and Allied Health Literature (CINAHL), Scopus, and Measure DHS. The databases were searched for published articles from January 1990 to December 2020. Results: Among the 1154 studies identified, only thirteen (13) studies met the study’s inclusion criteria. Narrative syntheses were performed on the selected papers to synchronize the various predictors identified. Factors ranging from child-related (age, birth order and use of a bed net), parental/household-related (maternal age and education status, household wealth index) and community-related variables (community wealth status, free bed net distribution) were some of the identified significant predictors. Conclusions: It is timely to have a synthesis of predictors that influence the malaria status of children under-five in SSA. The outcome of the review will increase the knowledge of the epidemiology of morbidity that will form the basis for designing efficient and cost-effective distribution of palliatives and control of malaria in SSA.

## 1. Introduction

Malaria has remained one of the major global public health challenges of the last two decades, especially in low and medium-income countries, putting almost half of the world population at risk of infection [1]. In 2015, over 214 million estimated cases of malaria were reported, with over 450,000 deaths worldwide [2,3]. Surprisingly, in 2018, the number of malaria cases had risen to over 300 million worldwide [4], and in 2019 the estimated malaria cases from 87 endemic nations were approximately 230 million [3]. It is on record that the world experienced a steady decline in the estimated number of deaths from malaria cases from over 730,000 in 2000 to over 400,000 in 2019 [3,5]. In 2017, there were an estimated 430,000 deaths; Sub-Saharan Africa (SSA) contributed over 90% of the global malaria deaths, with over 260,000 being children under five years of age, translating into one child dying every two minutes [2,6]. By 2019, under five years, deaths from malaria were over 60% of the total estimated deaths [3]. It is worth noting that in the last two decades, many programs and strategies have been implemented to control malaria both at the global and country levels, which has resulted in the prevention of over 6 million deaths between 2000 and 2015 in SSA [4].

Deaths and the burden of malaria cases among children under five years old vary across the various countries in SSA. Malaria alone contributes to more than 30 percent of under-fives mortality in Nigeria [5]. With over 51 million cases and 200,000 deaths annually, Nigeria has become the most malaria burdened nation in the world, with more than 30% of child mortality as a result of malaria cases [2,5]. In Tanzania, malaria is responsible for more than 10% of under-five deaths and is the second largest contributor to childhood morbidity and mortality [7] and is the leading cause of death in Mozambique, accounting for over 30% of all deaths [8]. Ethiopia, on the other hand, is recorded amongst the countries with the highest under-five mortality in SSA [9], with almost a quarter of the country being malaria-endemic, such that a greater proportion are exposed to malarial infection [4]. In 2016, Cameroon contributed about 3% of the total number of global deaths from malaria-related cases, and most of these deaths are among under-five-year-old children [10].

Apart from SSA, some other regions of the world contribute to the global burden of malaria. For instance, in 2019, out of the 107 malaria-endemic countries in the world, the Southeast Asia region had nine countries [3], making the area second only to Africa in terms of estimated malaria cases [11]. The South Asian region records between 90 and 167 million malaria cases, with over 125,000 deaths per annum [3,12]. Bangladesh is one of the four malaria-endemic countries in Southeast Asia [11], with over 17 million people at risk of malarial infection [13].

*Plasmodium falciparum* and *Plasmodium vivax* are the two most predominant malaria parasites causing malaria, with 60% and 40%, respectively, of cases in Ethiopia [14]. In India, this is estimated to be in the ratio 10:7 [12]. The reverse is the case for Brazil, where more than 60% of the 170 million people in the American region at risk of malaria cases reside, and over 70% of the cases are traceable to pv, with pf contributing more than 25% [15]. However, *Plasmodium falciparum* alone accounts for more than 95% of malaria cases in Nigeria [5,16].

Researchers have attributed the prevalence of malaria in SSA to several factors, which include medical conditions, environmental factors/seasonal influences and human status (such as age, gender, pregnancy, blood group and rhesus factors, among others), socioeconomic, demographic and area-related characteristics [4,17]. Malaria infection is said to be more prevalent in rural areas than in urban centers [14]. Until recently, malaria was believed to be a rural area disease because the transmitting vectors are said to breed more in the rural areas [14]. On a contrary note, Baragatti et al., 2009 [18] observed that malaria had remained a serious public health concern in urban areas. This is not unconnected with the general belief that developing urban centers will reduce the transmission of malaria infection [14]. Unfortunately, this is not the situation for most African countries with limited resources to provide adequate infrastructure amenities that cope with the rate of urbanization experienced, resulting in poor housing, sanitation and drainage systems, which could increase the vector breeding and human contacts [14]. Reports from studies on gender differences have also found mixed conclusions. For instance, a higher prevalence of malaria among boys than among girls has been reported in Oladeide et al. [19], while another study reported a higher prevalence among females than males [20]. The occurrence of mosquitoes, the vector for malaria, appears to be higher during the wet season than in the dry season [17]. However, the transmission rate of malaria is relatively higher in hotter regions, but with mountainous areas providing protection from transmission [8].

In Nigeria, for instance, as in most SSA, the need to measure the impact of the national malaria strategic plan (NMSP), 2014–2020, to reduce malaria-related mortality to zero by 2020 has resulted in a rise in the number of aged related studies on malaria [5]. As much as these studies are essential towards evidence-based healthcare decisions on malaria fever control in Nigeria and SSA, much more critical is the scoping of these studies. This has not been done, especially concerning determinants of the prevalence of malaria among children under five years in SSA. Therefore, this study aims to bridge this knowledge gap.

The Purpose of the Scoping Review

This scoping review aimed to find and evaluate the studies that describe the association between the socioeconomic, demographic, and contextual factors and the prevalence of malaria fever among children under five years of age in Sub-Saharan Africa.

## 2. Materials and Methods

### 2.1. Criteria for Inclusion and Exclusion of Studies

The scoping review followed the preferred reporting items for systematic reviews and meta-analyses extension for scoping reviews (PRISMA-ScR) checklists [21,22]. The review question was in line with the population, interventions, comparators, outcomes, timing and study design (PICOTS).

The population of the study was any child under five years in SSA countries, irrespective of their gender.

The exposures include socioeconomic, demographic, and contextual predictors. These interventions were either classified as child-related, parental/household-related or community-related. Where the study reported both adjusted and crude effect sizes, the adjusted was selected.

The comparator was between the children under five years in SSA that had malaria infection versus those that did not have malaria infection.

The outcome variable was the malaria status. This review considered studies on socioeconomic, demographic, and environmental determinants of malaria fever among children under five of both sexes that used standard testing procedures in identifying malaria fever status. The usual method of testing for the presence of *Plasmodium* parasites was by measuring the axillary temperature of 37.5 °C [23], and, carry out a microscopic examination of thick and thin blood smears that were positive with several asexual parasites per 200 white blood cells, while if white blood cells had a count of 8000 cells/μL [24]. In addition, studies, which identified malaria status through the rapid diagnostic test (RDT) and polymerase chain reaction (PCR) were included.

Studies and articles written in English and published between 1 January 1990 and 31 December 2020 were included.

The study design covers all observational studies (cross-sectional and cohort studies).

### 2.2. Search Strategy

The search strategy was first carried out in PubMed with the following terms as displayed in Table 1 and combined with appropriate Boolean connectors.

### 2.3. Sources of Information

The online databases for literature search using the search terms of PICOTS produced the following results as displayed in Figure 1. The results identified a total of 1154 publications. PubMed (867 results), Cumulative Index to Nursing and Allied Health Literature (CINAHL) (14), Scopus (0) MEDLINE (WOS) (122), Measure DHS (154). The databases were searched for published works from January 1990 to December 2020. The searches were done on 13 and 14 December 2020.

### 2.4. Study Selection

The selection processes followed the PRISMA-ScR recommendation [25]: identify all potential papers from databases and other sources; screen to remove duplicate publications; exclude those that did not meet up with inclusion criteria and inclusion for the review.

### 2.5. Data Selection Process

The reviewer PEO extracted relevant data from the selected studies into a Microsoft Excel spreadsheet (Microsoft Corporation, Washington, DC, USA); the design for this extraction was verified by the supervisory team.

## 3. The Results

### 3.1. Description of Study Records

This review aimed to synthesize evidence from published articles describing the determinants (socioeconomic, demographic and contextual) of the malaria status of children under five years in SSA between January 1990 and December 2020. The flowchart diagram in Figure 1 shows the selection of studies included for review. A total of 1157 records were identified from all the databases consulted. Ninety-three (93) were retained for full-text examination, and only thirteen (13) unique publications (with 18 country-specific studies) met the inclusion criteria and were examined for this review.

#### Study characteristics

Table 2 describes the characteristics of the 13 extracted papers considered for this review. The information includes the author’s name and date of publication, the title of the paper, study location, survey type, target sample, the prevalence of the outcome, sample size, statistical methods, and software used for computation. The prevalence of malaria in children under five, as reported in the papers reviewed, ranged from 18% in Tanzania to 39% in Uganda, as reported in Njau et al. (2013) [26]. The mean sample size per study was 30,775 participants.

Figure 2 displays charts A–G and describes the various study characteristics of the included papers. Chart A describes that the publication years for the included studies were between 2013 and 2020. In 2016 there were no publications extracted. The 2018 period had the highest number of publications (23%), while 2013 and 2019 had one publication each (representing 7.7%), and the remaining years had 15% each. In terms of the study setting (chart B), the number of studies from multi-countries (33%) was more than every other country-specific study. Nigeria, Malawi, Tanzania and Uganda had two studies each (11%), and the remaining countries had one study each.

With more than 70% of the number of country-specific studies, the malaria indicator survey (MIS) was the most used survey among the included studies (chart C), followed by multi surveys (16%) and demographic and health surveys (11%). Likewise, chart (D) indicates that multivariate logistic regression methods (46%) were the most used statistical method. Furthermore, with 38%, STATA was the most preferred statistical software used in the selected publications. This was followed by SAS, SPSS and combined software applications. The least popular was JMP (7.6%). Funding sources (chart F) show that multiple sources of funding were the highest at (38%), while no funding was reported in 30% of the studies, and two of the studies did not disclose any funding source. Three diagnostic methods were reported to detect malaria infection status in children under five years. Among the studies included were rapid diagnostic test (RDT), microscopic test (of thin and thick blood smear), and polymerase chain reaction (PCR). Chart (G) shows that RDT and the combination of RDT and MT were the most popular, and one study reported using PCR.

### 3.2. Data Synthesis Method

The narrative/tabulation syntheses of the outcomes/findings from multiple reports for this review following the narrative guidelines given in Popay et al. [38] were used. The objective of the review was to collate pieces of evidence on the association between the socioeconomic, demographic, and contextual indicators on malaria fever among children under five in SSA from 1990 to 2020. The results were appraised using the narration of the descriptive statistics and odds of the likelihood of the risk factors and the outcome variable (malaria status).

#### Predictors associated with Malaria Status

The socioeconomic, demographic, and contextual determinants of malaria status among children under five in SSA were grouped into child-related variables, mother or caregiver-related variables, household-related variables, and environmental or Area-related variables and interaction terms. These variables include age of the child, weight, anemia status, birth order status; maternal age and education status, parent’s knowledge, attitude and practices of some basic facts about malaria fever; the type of material used to construct the building, distance from a health facility and cluster altitude as factors identified that are associated with malaria status among children under five years in SSA. A factor was considered statistically significant concerning what each paper considered as the *p*-value cut-off (0.01, or 0.05, or 0.001). In a situation where the factor was classified into different categories or dummies, the factor was labeled as statistically significant if at least one of the categories or dummies compared to the reference category was statistically significant.

##### Child-Related Variables

Table 3 shows the evidence found on child-related variables. This study revealed the role that the age of the child plays in the tendency for the child to be infected with malaria parasites. Eleven (11) of the country-specific studies investigated a child’s age being under five years as a predictor of their potential malaria status. Nine of the studies found that the child’s age in at least one of the age groups was significantly associated with the prevalence of malaria among under five years in SSA. In most of the studies, it was found that as the child’s age increases, the odds of contracting malaria fever also increase [24,33,35,37,39]. However, Semakula et al. [32] in their multi-country study found no statistical significance in Tanzania (OR: 1.26, CI: 0.94–1.70, *p* = 0.128) and Burundi (OR: 0.79 CI: 0.60–1.05, *p* = 0.108), but found significant effect in Malawi (OR: 1.85 CI: 1.33–2.56, *p* < 0.001] and Liberia (OR: 2.10 CI: 1.59–2.80, *p* < 0.001]. Three studies investigated the sex of a child as a predictor of the prevalence of malaria among under five years old children in SSA. Surprisingly, these three studies ((OR: 0.96 CI: 0.91–1.02, *p* = 0.18) [39], (OR: 0.927, *p* = 0.2627) [35], and (OR: 1.04 CI: 0.82–1.32, *p* = 0.764)) [24] found no statistically significant effect. Also, three studies that explored the effect of birth order found statistically significant effects (OR: 1.03 CI: 1.01–1.06, *p* = 0.011) [27], when the child was second-order compared to the 1st child born: (OR: 1.43 CI: 1.04–1.96, *p* = 0.03) [28] and [marginal effect: 0.045, *p* < 0.01) [31].

Contrary to expectation, whether or not a child slept under a long-lasting insecticide-treated net was reported in five out of nine studies (OR: 0.93 CI: 0.84–1.02, *p* = 0.13) [27]; ((in Malawi study), OR: 0.88 CI: 0.73–1.07, *p* = 0.202) [32]; ((in Liberia study), OR: 0.99 CI: 0.85–1.17, *p* = 0.945) [32]; OR: 0.93 CI: 0.81–1.07] [33]; (OR: 1.5 CI: 0.9–2.4 *p* = 0.146) [37], not to be a significant predictor of malaria infection among children under five years in SSA.

##### Maternal-Related Variables

Table 4 describes the various factors predicting the likelihood that a child would contract malaria fever at the maternal-related-variable level. Out of three studies that analyzed maternal age as a predictor of contracting malaria fever among children under-fives in SSA, two studies (Njau et al., 2014 and Siri et al., 2014) [31,33] found no statistically significant effect. While, Berendsen et al., 2019 [27] found a statistically significant effect (OR: 0.99 CI: 0.98–0.99, *p* = 0.00047) of maternal age [27]. However, Zgambo et al., 2017 [37] did not find any statistically significant effect of maternal education on the likelihood of malaria infection among children under five years in SSA.

In addition, maternal knowledge of malaria fever was found to be a statistically significant predictor of under-five malaria cases in SSA. Children whose mothers showed having knowledge of malaria fever were less likely to be infested with malaria *parasitemia* (β: −0.013, *p* < 0.01) [31] and (OR: 0.78 CI: 0.62–0.99, *p* = 0.037) [24].

##### Household-Related Variables

Table 5 describes the distribution of significant effects of household-related variables on the likelihood of developing malaria infections among children under five years in SSA. The most widely assessed household-related predictors are household socioeconomic status (designated as household wealth), place of residence (whether urban or rural area), Household size, improved water source and improved toilet facilities. All eleven country-specific studies that investigated household wealth as a predictor found at least one of the categories being a statistically significant predictor of malaria status. The higher the household wealth quintile, the less likely it that the child in the household would contract malaria fever. The thirteen country-specific studies that found a statistically significant effect of the place of residence all reported that it was more harmful to a child under five years in rural SSA than in urban areas in contracting malaria fever. Though Wanzira et al., 2017 [24] and Zgambo et al. *2017* [37] found no statistically significant effect of place of residence, yet they reported a more protective effect for urban children than rural children (OR: 1.74 CI: 0.92–3.29, *p* = 0.089) [24], (OR: 2.3 CI: 0.9–6.0, *p* = 0.075) [37].

It is worthy of note that access to mass media, number of rooms in the household and type of wall material were found not to be statistically significant predictors of malaria fever among children under-five in SSA [26,35]. The variations in household ownership of livestock were a statistically significant predictor of malaria status in children under five years in SSA. Semakula et al., 2015 [32] reported consistent findings in their four country-specific studies that a child from a household that owns cattle has a lower odd of contracting malaria parasitemia than a child from a household without livestock (Tanzania (OR: 0.55 CI: 0.45–0.67, *p* < 0.001); Burundi (OR: 0.51 CI: 0.40–0.65, *p* < 0.001); Malawi (OR: 0.54 CI: 0.35–0.83, *p* < 0.001); Liberia (OR: 0.74 CI: 0.55–1.00, *p* < 0.05)).

##### Environmental/Area-related Variables

In consideration of environmental-related predictors, three variables (regional variations, malaria endemicity, and community free bed net distribution) were attractive for investigation among the included studies. Table 6 reports that Njau et al., 2013 [26] found that the predicted marginal effects (ME) of malaria-endemic areas for malaria fever in Angola, Tanzania and Uganda were significantly ME: 0.01 (*p* < 0.10), ME: 0.095 (*p* < 0.05) and ME: 0.288 (*p* < 0.01) points, respectively. Additionally, the same authors [26] reported an insignificant increase in the predicted marginal effects of 25.1% points for free bed net in the community among malaria positive children in Angola, but a significant reduction of 1.5% (*p* < 0.1) and 8.2% (*p* < 0.05) in Tanzania and Uganda, respectively.

Also, significant regional variations were reported across the six geopolitical zones of Nigeria. Morakinyo et al., 2018 [30], which found reduced odds of malaria infections among children 6–59 months in North Central (OR: 0.61 CI: 0.47–0.79, *p* < 0.01); North East (OR: 0.35 CI: 0.27–0.46, *p* < 0.01); North West (OR: 0.49 CI: 0.37–0.64, *p* < 0.01); South East (OR: 0.59 CI: 0.44–0.79, *p* < 0.01); South-South (OR: 0.42 CI: 0.31–0.55, *p* < 0.01), when compared with children from South West. Contrary to Morakinyo et al., 2018 [30] report on Nigeria study, Ugwu et al., 2018 [35] found insignificant effects on regional variations in South East, South-South, South West, and North Central when compared with North West, but found significant odd of malaria-positive cases among 6–59 months in North East (OR: 0.3059, *p* = 0.015) when compared with north West.

##### Interactions-related Variables

Interaction-related predictors were reported by two papers in four country-specific studies (Table 7). Njau et al., 2013 [26] reported a significant decrease of 4.6% (*p* < 0.05) points in the predicted marginal effects among malaria-positive children in Angola with respect to interaction terms of free bed net and wealth status, but found an insignificant reduction of 0.9% and 6.4% in Tanzania and Uganda, respectively.

Ugwu et al., 2018 [35] found significant interaction effects of wealth index (poorest, poorer, middle, richer and richest) and place of residence (rural or urban). In consideration of the report, the middle and richer household group in the rural area (OR: 0.448 CI: 0.2197–0.9124, *p* = 0.027 and OR: 0.417 CI: 0.2213–0.7871, *p* = 0.007) displayed a higher odd of malaria-positive than the poorest and poorer household group in the rural area (OR: 0.3567 CI: 0.1319–0.0429, *p* = 0.0429 and OR: 0.2770 CI: 0.1149–0.6677, *p* = 0.004) using richest and urban as a reference category. There were no significant interaction effects of region and place of residence on the odds of contracting malaria parasitemia among children 6–59 months in Nigeria.

## 4. Discussion

This study aimed to conduct a scoping review of the predictors that affect the malaria status of children under five years in SSA. The review found thirteen studies that identified factors associated with malaria fever among children under five in SSA. Though the search strategies covered the period from 1990 to 2020, the distribution of the publication years shows that papers conducted on predictors affecting the occurrence of malaria among children under-five in SSA were carried out in the last decade. All the publications (meeting the inclusion criteria) were from 2013 to 2020, and no publications in 2016 or 2012 and earlier were found. This may relate to the fact that the data set from nationally representative individual and household surveys in SSA, such as from malaria indicator surveys (MIS) designed a stand-alone survey [40] for use in areas where DHS and MCIS have not been used [41] were not often available until around 2012 and beyond [42]. Most of the MIS datasets collected from 2005 to 2012 remained unavailable within the period [42]. Furthermore, this review shows that the countries of the study were more concentrated in Southern and Central Africa, with just two recorded in West Africa. The reasons for this disparity are not clear. However, from a UNICEF report, the reduction in changes in the percentage of under-five mortality resulting from malaria between 2000 and 2017 was more drastic among the countries in Southern and Central Africa [6]. In addition, the regional disparity in the number of studies may be related to the fact that it is much easier for researchers to secure funding for their studies in Southern and Central Africa than in West African countries. There were more studies from the malaria indicator surveys (MIS) data set than from the Demographic and health surveys (DHS) data set. These differences are likely related to the fact that the timing of MIS is usually in the season where malaria infections are high [43,44], and it includes the use of biomarkers on the field and laboratory [44]. These reasons notwithstanding, technical assistance for both DHSs and MISs was provided for by DHS. In recent times, some country’s surveys combined both surveys into one. For instance, Nigeria conducted Nigeria’s malaria indicator survey (NMIS) in 2010 and 2015, conducted Nigeria demographic and health survey (NDHS) in 2008 and 2013 separately, but NDHS 2018 was a combination of both NDHS and NMIS [45].

From this review, the most vulnerable, in children under five in terms of age, are those between 2 to 5 years [24,30,33,35,37]. This finding agrees with report from another study [46]. The reasons may not be unconnected with the fact that most families, when they have new-born, intra-family attention and use of resources are shifted to the new-born. As revealed by this review, another factor of importance is the significance of the increase in the maternal/caregiver educational status has on protecting the child from having malaria parasitemia [24,27,28,31,33,35,37]. This is also in line with the findings in Mehretie et al. [47]. One of the pathways in which this can affect the malaria status of under-5 is through adequate knowledge of malaria symptoms, prompt response to seek healthcare attention [24,31].

## 5. Strengths and Limitations

There are several strengths identified in this scoping review. (i) This is the first scoping review that was carried out on predictors affecting the malaria status of under five years in SSA that used classical statistical regression methods on data from a secondary analysis of nationally representative surveys. (ii) The review was rigorous with intense supervision from the team that cuts across two institutions.

It is acknowledged that this review has some limitations. (i) All the studies considered in this review are the secondary analysis of nationally representative cross-sectional surveys. Causal effects are not established in the studies, and cross-sectional studies, which are carried out for a time point, cannot determine trends [48]. (ii) Very few studies reviewed in this project considered the contextual factors that may be associated with the malaria status of under-fives with appropriate statistical technique; this may have reduced the reliability of the results attained. Studies that can investigate the contextual factors related to malaria fever among children under five in SSA countries are urgently needed. (ii) Only thirteen studies were identified that met the inclusion criteria. The study may not have successfully identified all the papers as the only considered studies were those that were written in the English language [49]. (iii) In view of the scoping review study design applied [50], we also acknowledged that there was no publication bias and quality of study assessment done. (iv) Only studies that applied frequentist statistical methods were included. Therefore, the exclusion of studies that applied Bayesian statistical methods could have resulted in limitations in the findings. (v) The intervention terms were omitted in the search; this may have excluded some potential studies.

## 6. Future Work

There are a few areas not covered in the papers included in this review that require future investigations. Considering the limitations stated above, a scoping review that will take care of them should be the basis for future study. Malaria infection in children is comorbid and, as such, may have overlapping associative predictors. Studies that could explore this area are a potential study area for the future.

## 7. Conclusions

SSA is one of the high endemic malaria regions in the world, with a high mortality rate resulting from malaria morbidity. There is a more significant commitment on the part of government and partners to ensure that morbidity and mortality resulting from malaria fever in some countries is reduced to zero by the end of 2020 [5]. The target year is now passed, but it does not seem to have been achieved. The knowledge derived from a careful analysis of these many factors contributing to the rising burden could be used to fast track appropriate intervention mix as they become available [51]. For instance, children delivered in health facilities have a reduced risk of malaria infections when compared with those, who are delivered at home, so a child’s place of delivery, child’s anemia status, vaccination status, access to insecticide-treated bed net, maternal education status, number of births within five years, duration of breastfeeding, improved water source, availability of electricity, a constant residual spray of houses and environment, and community distance to healthcare facilities are areas that need government attention through policymaking and implementations to reduce malaria infection rates among children under five years. Generally, some population settings, especially the children under five, are more at risk than others, where over 70% of mortality from malaria occurs [52], and measures are needed to protect these vulnerable groups [53]. However, vector control (such as insecticide-treated mosquito nets, drug treatments and indoor residual spraying) is one of the main approaches most SSA governments have adopted to prevent and reduce the spread of malaria [52,53,54] caused through mosquito bites [55]. Inadequate knowledge of how the individual and contextual factors are associated with malaria contraction may jeopardize governments’ ability to eliminate the malaria parasite [54].

## Figures and Tables

**Figure 1 ijerph-18-02119-f001:**
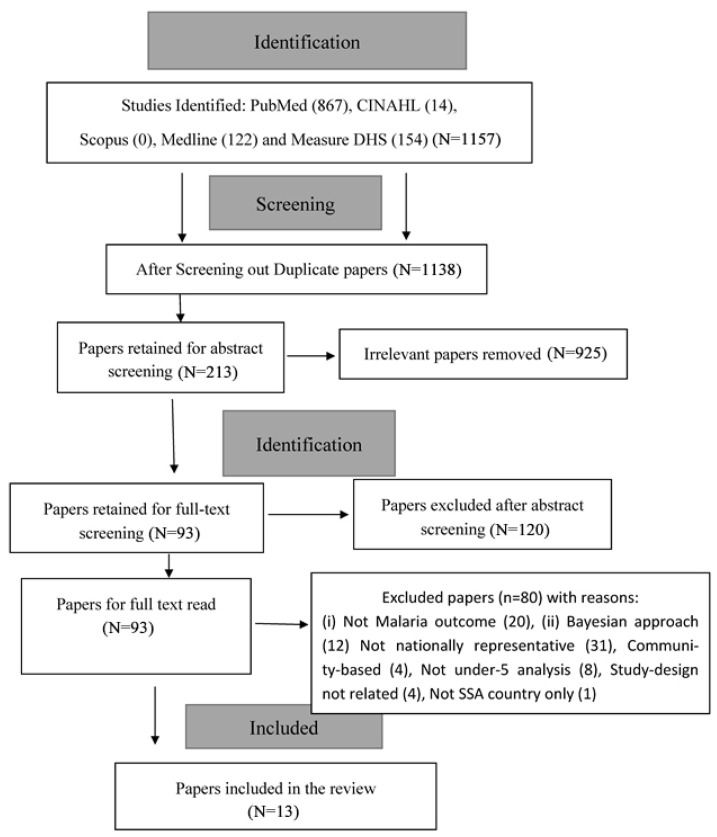
Flowchart of inclusion of studies for malaria review.

**Figure 2 ijerph-18-02119-f002:**
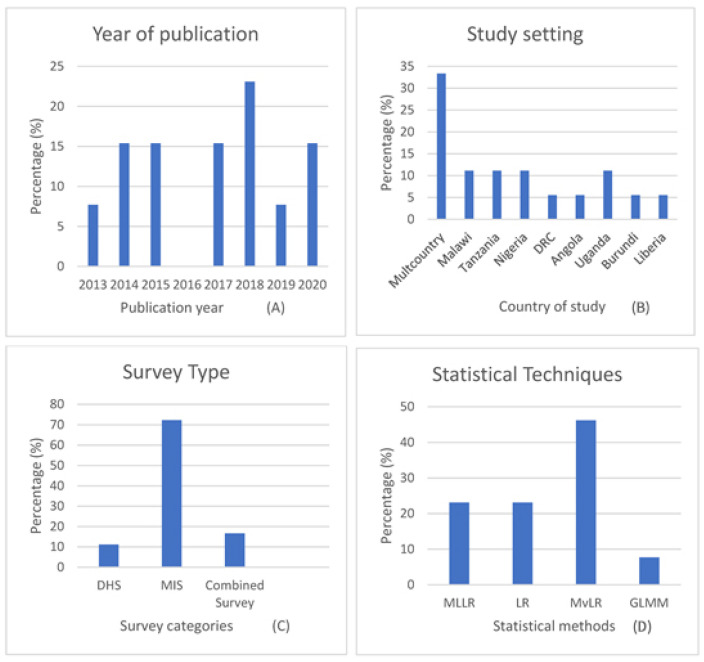
Charts (**A**–**G**) represent the distributions of the characteristics of included studies.

**Table 1 ijerph-18-02119-t001:** Search terms combinations.

S/N	Search Terms
1	demographic health survey OR AIDS indicator survey OR malaria indicator survey OR multiple indicator cluster surveys OR health survey OR MIS OR DHS
2	sub-Sahara Africa OR SSA
3	logistic regression OR multilevel regression OR multinomial logistic OR random-effects OR hierarchical OR fixed effects OR Linear regression
4	Malaria OR fever OR *plasmodium falciparum* OR *P. malariae* OR *P. ovale* OR *P. vivax*)
5	1 AND 2 AND 3 AND 4

**Table 2 ijerph-18-02119-t002:** Characteristics of the included studies.

Authors and Dates	Titles	Country	Survey *	Target Population	Prevalence n (%)	Participants (Sample Size)	Malaria Diagnostic Method **	Methods	Software	Funding Source
Berendsen et al., 2019 [27]	BCG vaccination is associated with reduced malaria prevalence in children under the age of 5 years in Sub-Sahara Africa	Multi-country (13 SSA)	DHS	Under 5 years	12,325 (36)	34,205	RDT	Multilevel logistic regression (MLLR)	SPSS, STATA, MLWin	Multiple source
Chitunhu et al., 2015 [28]	Direct and indirect determinants of childhood malaria morbidity in Malawi: a survey cross-sectional analysis based on malaria indicator survey data for 2012	Malawi	MIS	Under 5 years	367 (27.7)	1375	MT	Logistic regression (LR)	STATA	Institution-based
Levitz et al., 2018 [29]	Effect of individual and community-level bed net usage on malaria prevalence among under-fives in the Democratic Republic of Congo	Democratic Republic of Congo (DRC)	DHS	Under 5 years	2191 (37.4)	5857	Others (PCR)	Multilevel logistic regression (MLLR)	SAS	Multiple sources
Morakinyo et al., 2018 [30]	Housing type and risk of malaria among under-five children in Nigeria: evidence from the malaria indicator survey	Nigeria	MIS	6–59 months		6991	RDT and MT	Logistic regression (LR)	STATA	No funding
Njau et al., 2013 [26]	Exploring the impact of targeted distribution of free bed nets on households bed net ownership, socioeconomic disparities and childhood malaria infection rates: analysis of national malaria survey data from three sub-Saharan Africa countries	Angola, Tanzania and Uganda	MIS	Under 5 years	214 (20) 895 (39) 782 (18)	1125 3109 1954	RDT and MT	Multilevel logistic regression (MLLR)	STATA	Multiple source
Njau et al., 2014 [31]	Investigating the Important Correlates of Maternal Education and Childhood Malaria Infections	Angola, Tanzania and Uganda (Pooled)	MIS	Under 5 years	-	1390 5975 2997	RDT	Multivariate logistic regression (MvLR)	STATA	Not reported
Semakula et al., 2015 [32]	Potential of household environmental resources and practices in eliminating residual malaria transmission: a case study of Tanzania, Burundi, Malawi and Liberia	Tanzania, Burundi, Malawi and Liberia	MIS	Under 5 years	-	7695 3750 2115 3187	RDT	Multivariate logistic regression (MvLR)	JMP 10	Multiple source
Siri 2014 [33]	Independent Associations of Maternal Education and Household Wealth with Malaria Risk in Children	Multi-country (pooled)		Under 5 years	-	24,043	-	Multivariate logistic regression (MvLR)	SAS	Institution-based
Tusting et al., 2020 [34]	Housing and child health in sub-Saharan Africa: A cross-sectional analysis	Multi-country (pooled)	Multiple surveys	Under 5 years	40,178 (21)	188,651	RDT and MT	Conditional logistic regression (LR)	STATA and R	Multiple source
Ugwu and Zewotir, 2018 [35]	Using mixed effects logistic regression models for complex survey data on malaria rapid diagnostic test results	Nigeria	MIS	6–59 months	-	5236	RDT	Generalized linear mixed model (GLMM)	SAS	No funding
Wanzira et al., 2017 [24]	Factors associated with malaria parasitaemia among children under 5 years in Uganda: a secondary data analysis of the 2014 Malaria Indicator Survey dataset	Uganda	MIS	Under 5 years	938 (19.04)	4930	MT	Multivariate logistic regression (MvLR)	STATA	no funding
Yang et al., 2020 [36]	Drinking water and sanitation conditions are associated with the risk of malaria among children under five-year-old in sub-Saharan Africa: A logistic regression model analysis of national survey data	Multi-country (pooled)	Multiple surveys	Under 5 years	40,217 (18.8)	213,920	RDT and MT	Multivariate logistic regression (MvLR)	SPSS	not reported
Zgambo et al., 2017 [37]	Prevalence and factors associated with malaria parasitaemia in children under the age of five years in Malawi: A comparison study of the 2012 and 2014 Malaria Indicator Surveys (MISs)	Malawi	MIS	Under 5 years	636 (33)	1928	MT	Multivariate logistic regression (MvLR)	SPSS	no funding

* Surveys: MIS = malaria indicator survey, DHS = demographic and health survey; ** RDT = rapid diagnostic test, MT = microscopic test.

**Table 3 ijerph-18-02119-t003:** Association between child-related variables and malaria status.

S/N	Variables	Significance Levels	Number of Studies	Association Effect (95% CI)
1	Age of the child	S:	9	Increased significant factors (ISF) OR: 1.05 (1.04–1.06) [27] OR: 1.03 (1.02, 1.04) [28] 7–23: OR: 2.29 (1.21–4.34), 24–59: OR: 5.67 (3.01–10.70) [30] OR: 1.85 (1.33–2.56) [32] OR: 2.10 (1.59–2.80) [32] 6–11: OR: 2.22 (1.88, 2.62); 12–23: OR: 3.70 (3.12, 4.37) 24–35: OR: 5.00 (4.25, 5.87) [33] 13–24: OR: 1.7039 (1.34–2.16); 25–36: OR: 2.624 (2.06–3.33); 37–48: OR: 3.591 (2.82–4.55); 49–59: OR: 4.97 (3.888–6.38) [35] 7–12: OR: 1.62 (1.04–2.52); 13–24: OR: 2.20 (1.47–3.29); 25–36: OR: 3.47 (2.32–5.20); 37–48: OR: 3.69 (2.47–5.50); 49–59: OR: 4.01 (2.57–6.45) [24] 24–35: OR: 1.5 (1.0–2.5) ≥48: OR: 2.2 (1.4–3.5) [37] decreased significant factors (DSF) 36 month+ OR: 0.80 (0.72, 0.88) [33]
		NS:	2 [32]	
2	Vaccination status	S:	1	DSF: OR: 0.88 (0.82 to 0.94) [27]
		NS:	-	
3	Preceding birth interval	S:	1	ISF: OR: 1.00 (1.00 to 1.00) [27]
		NS:	-	
4	Birth order	S:	3 [27,28,31]	ISF: OR: 1.03 (1.01–1.06) [27] Second: OR: 1.43 (1.04, 1.96) [28] β: 0.045 [31]
		NS:	-	
5	Breastfeeding status	S:	1	DSF: currently: 0.85 (0.73–0.99) [27]
		NS:	-	
6	Fever in the last 2 weeks	S:	1	ISF: OR: 1.967 (1.71–2.26) [35]
		NS:	-	
7	Anemic	S:	2	ISF: OR: 2.982 (2.54–3.49) [35] DSF: OR: 0.95 (0.94, 0.96) [28]
		NS:	-	
8	Place of delivery	S:	1	DSF: public: 0.85 (0.78 to 0.92); private: 0.78 (0.70 to 0.87) [27]
		NS:	-	
9	Child slept under a mosquito bed net	S:	4	ISF: OR: 1.21 (1.08–1.36) [30] OR: 1.47 (1.16–1.89) [32] DSF: OR: 0.77 (0.60, 0.99) [28] OR:0.65 (0.56–0.77) [32]
		NS:	5 [27,32,33,37]	

OR: odds ratio, ME: marginal effect, β: coefficient estimate, S: significant, NS: not significant, ISF: increased significant factors, DSF: decreased significant factors.

**Table 4 ijerph-18-02119-t004:** Association between maternal-related variables and malaria status.

S/N	Variables	Significance Levels	Number of Country Studies	Association Effect (95% CI)
1	Maternal age	S:	1	DSF: OR: 0.99 (0.98 to 0.99) [27]
		NS:	2 [31,33]	
2	Maternal education status	S:	6	ISF: no Education: OR: 2.0454 (1.36–3.07); primary: OR: 1.5311 (1.03–2.28); secondary+: OR: 1.547 (1.07–2.23) [35] DSF: primary: OR: 0.91 (0.86 to 0.96); secondary+: OR: 0.73 (0.67 to 0.78) [27]. primary: OR: 0.53 (0.37, 0.76) [28] PS: β: −0.032; above primary: β: −0.047 [31] OR: 0.993 (0.990–0.996) [33] Primary: OR: 0.75 (0.59–0.96); secondary: OR: 0.61 (0.43–0.86); Tet: OR: 0.11 (0.02–0.53) [24]
		NS:	1 [37]	
3	Maternal body mass index	S:	1	DSF: OR: 0.97 (0.96–0.98) [27]
		NS:	-	
4	Maternal ante-natal care	S:	1	DSF: β: −0.029 [31]
		NS:	-	
5	Number of births in 5 years	S:	1	ISF: OR: 1.08 (1.03–1.13) [27]
		NS:	-	
6	Maternal knowledge of malaria fever	S:	2	ISF β: 0.013 [31] DSF: yes: OR: 0.78 (0.62–0.99) [36]
		NS:	-	
7	Number of children ever born	S:	1	ISF β: 0.003 [31]
		NS:	-	
8	Mother has access to phone	S:	1	DSF: β: −0.030 [31]
		NS:	-	

OR: odds ratio, ME: marginal effect, β: coefficient estimate, S: significant, NS: not significant, ISF: increased significant factors, DSF: decreased significant factors.

**Table 5 ijerph-18-02119-t005:** Association between household-related variables and malaria status.

S/N	Variables	Significance Levels	Number of Country Studies	Association Effect (95% CI)
1	Household wealth status	S:	11	ISF: international wealth index square: 1.00 (1.00 to 1.00) [27] poor: 5.51 (3.83–7.93) poorer: 5.15 (3.72–7.13) middle: 3.51 (2.64–4.65) richer: 1.89 (1.46–2.45) [30] poorest: OR: 3.5498 (1.508–8.35); poorer: OR: 5.6013 (2.69–11.63); middle: OR: 2.4569 (1.46–4.12); richer: OR: 1.8258 (1.24–2.67) [35]; poorest: OR: 4.7 (1.3–16.2) [37] DSF: OR: 0.95 (0.93–0.98) [28] ME: −0.034 (−0.1543– 0.0773) [26]; ME: −0.070 (−0.0943–0.0267) [26]; ME: −0.116 (−0.1876–−0.0583) [26] poor: β: −0.019 (0.017); less poor: β: −0.033 (0.018); middle: β: −0.065 (0.018); rich: β: −0.123 (0.019) [31] OR: 0.990 (0.987–0.992) [33] poorer: 0.70 (0.50–0.99); middle: 0.75 (0.50–1.12) 0.157; richer: OR: 0.40 (0.27–0.61); richest: OR: 0.17 (0.08–0.36) [24]
		NS:	-	
2	Place of residence	S:	13	ISF: rural: OR: 1.91 (1.63–2.25) [27] rural: OR: 1.83 (1.18–2.83) [28], rural: OR: 1.59 (1.33–1.89) [30], ME: 0.002 (0.0781–0.1228) ME: 0.055, CI: (0.0005–0.1097) [26] rural: β: 0.024 [31] rural: OR: 4.57 (1.86–11.25) [35] DSF: urban: OR: 0.94 (0.61–1.42) [32] OR: 0.26 (0.13–0.49) [32] urban: OR: 0.39 (0.25–0.60) [32] urban: OR: 0.72 (0.570.92) [32] urban: OR: 0.59 (0.50–0.71) [33]
		NS:	2 [24,37]	
3	Household had bed net	S:	4	DSF: ME: −0.055 (−0.1187–0.008) [26]; ME: −0.034 (−0.1233–0.0387) [26] ME: −0.098 (−0.0419–0.1494) [26] β: −0.076 [31]
		NS:	1 [37]	
4	Age of household head	S:	4	ISF: ME: 0.006 (−0.0004–0.0016) [26]; ME: 0.001 (−0.0005–0.0029) [26], OR: 1.019 (1.007–1.031) [35] DSF: ME: −0.009 (0.0012–0.0032) [26]
		NS:	-	
5	Insecticide residual spray	S:	2	DSF: OR: 0.37 (1.08–1.36) [30] OR: 0.23 (0.08–0.61) [24]
		NS:	2 [35,37]	
6	Household size	S:	7	ISF: OR: 1.03 (1.01–1.04) [27] ME: 0.015 (0.0021–0.0285) [26] ME: 0.004 (−0.0059–0.0050) [26] ME: 0.005 (−0.0163–0.0055) [26] β: 0.009 [31] OR: 1.46 (1.24–1.73) [33], OR: 1.108 (1.03–1.17) [35]
		NS:	-	
7	Number of under-5 in household	S:	3	ISF: ME: 0.049 (0.0331–0.6565) [26] DSF: ME: −0.025 (−0.1787–−0.0181) [26] ME: −0.044 (−0.0742–−0.0156) [26]
		NS:	1 [31]	
8	Source of water outside	S:	1	DSF: OR: 0.97 (0.96, 0.99) [28]
		NS:	1 [35]	
9	Improved water source	S:	5	ISF: borehole: OR: 1.50 (1.10–1.88); unprotected well: OR: 1.56 (1.29–1.88); protected well: OR: 2.19 (1.53–3.10); river/lakes: OR: 2.45 (1.81–3.31) [32] borehole: OR: 1.75 (0.61–0.93); protected well: OR: 1.44 (0.25–0.78) [32] borehole: OR: 1.19 (0.36–3.60); protected well: OR: 1.36 (1.041.78); unprotected spring: OR: 1.65 (1.012.71) 0.047; river/lakes: OR: 1.55 (1.12–2.16) [32] unprotected: OR: 1.17 (1.07, 1.27) [36] DSF: piped (yard): OR: 0.13 (0.03–0.32); public pipe: OR: 0.70 (0.51–0.95); private taps: OR: 0.62 (0.39–0.95) protected spring: OR: 0.78 (1.06–2.83) [32]. piped (yard): OR: 0.05 (0.00–0.58); public pipes: OR: 0.52 (1.25–1.84); private tap: OR: 0.23 (0.04–0.75) [32]; piped (yard): OR: 0.23 (0.12–0.43); public: OR: 0.33 (0.23–0.47) [32] public: OR: 0.27 (0.13–0.51) [32] piped: 0.52 (0.45–0.59) [36]
		NS:	2 [27,35]	
10	Improved toilet facility	S:	7	ISF: open toilet: OR: 1.35 (1.11–1.63) no toilet: OR: 3.57 (2.35–5.42); pit: OR: 1.30 (1.07–1.58) [32] no toilet: OR: 1.66 (1.20–2.30) [32] no toilet: OR: 1.24 (0.821.28 [32] no toilet: 1.635 (1.209–2.21) [35] no toilet: OR: 1.35 (1.24, 1.47) [36] DSF: medium-quality: OR: 0.85 (0.78 to 0.92) [27]; flush toilet: 0.40 (0.18–0.78) [32] flush toilet: OR: 0.04 (0.02–8.01) [32] flush toilet: 0.53 (0.390.73) [32]; flush toilet: OR: 0.51 (0.43, 0.61) [36]
		NS:	-	
11	Sex of household head	S:	1	DSF: male: ME: −0.029 (−0.0637–0.0049) [26]
		NS:	4 [26,31,32]	
12	Use biomass for cooking	S:	2	ISF: firewood: OR: 1.80 (1.23–2.68) [32] firewood: OR: 1.44 (0.98–2.16) [32] DSF: charcoal: OR: 0.58(0.38–0.85) [32]
		NS:	-	
13	Under 5 years child slept under bed net	S:	2	ISF: yes: OR: 1.33 (1.04–1.71) [24] DSF: OR: 0.83 (0.78–0.88) [34]
		NS:	1 [35]	
14	Household ownership of livestock	S:	4	ISF: goat: OR: 1.32 (1.09–1.60) [32] goat: 1.26 (1.07–1.48) OR: 1.17 (0.98–1.38) [32] DSF: cattle: OR: 0.55 (0.45–0.67) pigs: OR: 0.18 (0.09–0.33) [32] cattle: OR: 0.51 (0.40–0.65) [32] cattle: OR: 0.54 (0.35–0.83) cattle: OR: 0.74 (0.55 1.00) [32]
		NS:	-	
15	Improve building materials	S:	2	ISF: nothing improved: OR: 1.05 (1.02–1.12) [30]; OR: 0.88 (0.83–0.93) [34]
		NS:	1 [34]	
16	Household head education status	S:	2	ISF: ME: 0.027 (−0.0023–0.0567) [26] DSF: primary school+: β: −0.009 (0.004) [31]
		NS:	1 [26]	
18	Household connected electricity	S:	1	ISF: no: OR: 1.14 (0.88–1.48) [35]
		NS:	-	
19	Roofing material	S:	1	DSF: palm leaf: OR: 0.7171 [35]
		NS:	-	

OR: odds ratio, ME: marginal effect, β: coefficient estimate, S: significant, NS: not significant, ISF: increased significant factors, DSF: decreased significant factors.

**Table 6 ijerph-18-02119-t006:** Association between environmental-related variables and malaria status.

S/N	Variables	Significance Levels	Number of Country Studies	Association Effect (95% CI)
1	Community wealth status	S:	1	ISF: cluster level: OR: 0.984 (0.979, 0.988) [33]
		NS:	-	
2	Community distance to health facilities	S:	2	ISF: ME: 0.084 (0.0560–0.1128) [26] ME: 0.102 (0.0525–0.1521) [26]
		NS:	-	
3	Cluster altitude	S:	1	ISF: OR 1.0003 (0.991–1.1003) [35]
		NS:	1 [28]	
4	Community insecticide net use	S:	1	ISF: OR: 0.43 (0.27, 0.70) [29]
		NS:	-	
5	Regional variations	S:	3 [24,28,30]	
		NS:	1 [37]	
6	Malaria endemicity	S:	4	ISF: ME: 0.010 (−0.0778–0.0572) [26] ME: 0.095 (0.0357–0.1561) [26] ME: 0.288 (−0.5526–−0.0247) [26] high: β: 0.093 [31]
		NS:	-	
7	Free bed net in community	S:	3	ISF: ME: 0.251 (0.0226–0.4801) [26] DSF: ME: −0.015 (−0.0134–0.0405) [26] ME: −0.082 (0.1479–0.0494) [26]
		NS:	-	
	Country-specific	S:	1	ISF: Liberia: OR: 1.09 (0.95–1.24); Uganda: OR: 40.15 (29.74–54.20); Malawi: OR: 16.68 (12.38, 22.48); Senegal: OR: 1.01 (0.77, 1.32); Nigeria: OR: 31.91 (23.86, 42.67) [33] DSF: Rwanda: OR: 0.15 (0.10, 0.21); Tanzania: OR: 0.82 (0.63, 1.07); Madagascar: OR:0.73 (0.57, 0.94) [33]
		NS:	-	

OR: odds ratio, ME: marginal effect, β: coefficient estimate, S: significant, NS: not significant, ISF: increased significant factors, DSF: decreased significant factors.

**Table 7 ijerph-18-02119-t007:** Association between interaction-related variables and malaria status.

S/N	Variables	Significance Levels	Number of Country Studies	Association Effect (95% CI)
1	Free bed net/wealth status	S:	1	DSF: ME: −0.046 (−0.0668–0.1772) [26]
		NS:	2 [26]	
2	Wealth/place of residence	S:	1	DSF: poorest/rural: OR: 0.3567 (0.13–0.96); poorer/rural: OR: 0.2770 (0.11–0.66); middle/rural OR: 0.4477 (0.22–0.91); richer/rural: OR: 0.4174 (0.22–0.78) [35]
		NS:	-	
3	Number in household/age of household head	S:	1	DSF: OR: 0.9984 (0.997–0.999) [35]
		NS:	-	

OR: odds ratio, ME: marginal effect, β: coefficient estimate, S: significant, NS: not significant, ISF: increased significant factors, DSF: decreased significant factors.

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
