# Peer review of "A Scoping Review of Selected Studies on Predictor Variables Associated with the Malaria Status among Children under Five Years in Sub-Saharan Africa"

_ijerph, 2021, doi:10.3390/ijerph18042119_

Round 1

Reviewer 1 Report

It is a well-written manuscript, but it could be improved as follows:

  1. The inclusion criteria of the published study seems too tough, leading to only 13 studies included in this review.  Too fewer studies/manuscripts will make the review not representative.  It is suggested that the authors should make some adjustments for paper selection criteria to include more studies.  
  2. SSA is the target geographic regions of the review.  However, the authors could list the comparison between SSA and other regions such as south east Asia, south Asian where the malaria is also one of major health problems and there are many developing countries.
  3. Why children under age of five is specifically mentioned is this study?  If a research covering kids under age of 10, whether it is worthwhile including in this study?
  4. Whether the author could list the risk factors of malaria into doable and non-doable factors for government or medical task forces, which should be more helpful for decision making and resource allocations.

Author Response

Reviewer one

  1. The inclusion criteria of the published study seems too tough, leading to only 13 studies included in this review.  Too fewer studies/manuscripts will make the review not representative.  It is suggested that the authors should make some adjustments for paper selection criteria to include more studies

The study is part of a larger doctoral study involving two other outcomes (which are already published). For uniformity, the key characteristics for inclusion where adhered to. However, this limitation has been recognised in the paper.

  1. SSA is the target geographic regions of the review.  However, the authors could list the comparison between SSA and other regions such as south east Asia, south Asian where the malaria is also one of major health problems and there are many developing countries.

The statement of comparison with other regions has been inserted in lines 65 – 78

3.  Why children under age of five is specifically mentioned is this study?  If a research covering kids under age of 10, whether it is worthwhile including in this study?

The scope of the unit of analysis was under-5 years. However, if a research covered children under the age of 10 years, this will be included provided the analysis specifically reported for the segment of the age under five years differently from other age segment.

4.  Whether the author could list the risk factors of malaria into doable and non-doable factors for government or medical task forces, which should be more helpful for decision making and resource allocations

A short paragraph has been added to itemize the identified risk factors that government could make and implements policies to correct the imbalance toward reducing the rate of malaria infections

Reviewer 2 Report

A Scoping Review of Selected Studies on Risk Factors Associated with the Malaria Status among Children Under-five Years in Sub-Sahara Africa

In this review, as indicated by the title, the authors carried out a systematic revision and analyzed the risk factors associated with malaria fever/disease in children under-five years. In my opinion, the review concentrates relevant information for children under 5 years old and their risk to get malaria important as this age group is the most vulnerable to severe malaria and death. The review might be more concise, if the most striking and consistent results are shown in Tables or in the text, and avoid redundant information or that that does not address the objective. The conclusions must comprise the most striking finding from the analysis.

Some specific suggestions and comments

Fig. 2 it is not clear why the information of the 93 studies are shown if in Figure 1 authors mentioned that those studies did not meet the inclusion criteria. Any use of this?.

“Table 3. Association between child-related variables and malaria status.” it is hard to read. I suggest indicating the meaning of DRF and IRF on the table footnotes. What is the difference between this organization “OR:1.05 (1.04-1.06)[27]” and “IRF: Yes: OR:2.982 (2.54-3.49) [35]”. The “IRF: Yes:” is necessary? might be confusing. Some variables are not strictly related to the title e.g. Multiple births or Place of delivery. To be more concise delete from the table those results that are not very important and only not significant (e.g. Vitamin A, multiple births, and sex), they might be mentioned in the text, and help the readers to see the most important finding at a glance. Please revise. Some of those observations are similar to Table 5 and Table 6.

Table 4. The abbreviations or acronyms Pry, BMI, ANC, Pry, sec, Tet, should be indicated in the table footnotes.

Please verify P- values throughout the paper. It is hard to read some of them, e.g. (line 315) p<0.047, in my opinion, itt should be p<0.05 (if uniform), or is it p=0.047? because it is an exact value. Also, “(line 321-322) Angola, Tanzania and Uganda were significantly 1.0 (p<0.10), 9.5 (p<0.05) and 28.8 (p<0.01)”.. so, P-value: p<0.1 meaning? Everything downwards Include significant and no significant values (e.g. p=0.099 is not significant but below 0.05 would be).

In the discussion, the first large paragraph seems to be mentioned in the result section, it is redundant. I suggest deleting it. Genus and species should be in italics.

The first time a species is mentioned write complete genus, e,g, line 72, 77, 126, Table 1, Table 3 (S/N 1, 10.70 [32],

Uniform dates and periods, e.g. line 133 “1 January 1990 and 31 December 2020. “ vs line 145-146 “1 January 1990 to 28 th December 2020. The searches were done on the 13 th and 14 th December 2020.” And others Avoid redundancy: lines 53-55, revise it.

Revise the spaces to add or remove and errors in different pages e.g. lines 44, 110, Table 1, Figure 1, lines 182, 186, 223, 24, 296, etc.

Square brackets or parenthesis? If for references Square brackets are used, then do not use them for other purposes, line 305.

Line 130: Change “ul” to “µL”

Author Response

Reviewer two

Fig. 2 it is not clear why the information of the 93 studies are shown if in Figure 1 authors mentioned that those studies did not meet the inclusion criteria. Any use of this?.

The information on fig. 2 are not the 93 studies, but summary of some characteristics of the 13 unique papers included. The first paragraph of lines 20.. has been adjusted to make it clearer

“Table 3. Association between child-related variables and malaria status.” it is hard to read. I suggest indicating the meaning of DRF and IRF on the table footnotes. The meaning of IRF and DRF are indicated in foot notes of the relevant tables

What is the difference between this organization “OR:1.05 (1.04-1.06) [27]” and “IRF: Yes: OR:2.982 (2.54-3.49) [35]”. The “IRF: Yes:” is necessary? might be confusing. Corrected on the tables.

Some variables are not strictly related to the title e.g. Multiple births or Place of delivery.  These were as reported in the papers. They were considered as child-related (a child resulting from multiple birth as compared with single birth and a child delivered from health facility or at home) To be more concise delete from the table those results that are not very important and only not significant (e.g. Vitamin A, multiple births, and sex), they might be mentioned in the text, and help the readers to see the most important finding at a glance. These variables and similar ones have been removed from the tables. Please revise. Some of those observations are similar to Table 5 and Table 6.

Table 4. The abbreviations or acronyms Pry, BMI, ANC, Pry, sec, Tet, should be indicated in the table footnotes. These are corrected in the tables

Please verify P- values throughout the paper. It is hard to read some of them, e.g. (line 315) p<0.047, in my opinion, itt should be p<0.05 (if uniform), or is it p=0.047? because it is an exact value. Also, “(line 321-322) Angola, Tanzania and Uganda were significantly 1.0 (p<0.10), 9.5 (p<0.05) and 28.8 (p<0.01)”..

Table 6 reports that Njau et al 2013 [26] found that the predicted marginal effects of malaria-endemic areas for malaria fever in Angola, Tanzania and Uganda were significantly 1.0 (p<0.10), 9.5 (p<0.05) and 28.8 (p<0.01) percentage points, respectively.

The 1.0, 9.5 and 28.8 are percentage marginal effects as reported in the paper. The paper in question identified significant risk factors at different significance levels: 10%, 5% and 1%

However, we have corrected the statement to read in parts … were significantly ME: 0.01 (p<0.10), ME: 0.095 (p<0.05) and ME: 0.288 (p<0.01) percentage points

In the discussion, the first large paragraph seems to be mentioned in the result section, it is redundant. I suggest deleting it. Deleted as recommended. But the style we used here was similar to the two other similar papers already published by highlighting the major findings in the discussion section

Genus and species should be in italics Complied with.

Uniform dates and periods, e.g. line 133 “1 January 1990 and 31 December 2020. “ vs line 145-146 “1 January 1990 to 28 th December 2020. The searches were done on the 13 th and 14 th December 2020.” Corrected

And others Avoid redundancy: lines 53-55, revise it. I don’t understand why these are redundant. We tried to give situation report for some selected SSA specific countries such as Nigeria (the most malaria endemic nation in the region), Tanzania, Mozambique, Ethiopia and Cameroon (to mention a few)

Revise the spaces to add or remove and errors in different pages e.g. lines 44, 110, Table 1, Figure 1, lines 182, 186, 223, 24, 296, etc.

Line 44:  the word ‘estimated’ has been added as reported in ref [3] in table 3.1

Line 110, the period removed

Table 1, space added

…..

Square brackets or parenthesis? If for references Square brackets are used, then do not use them for other purposes, line 305.

This may not be a problem because MDPI normally distinguish the references square brackets from other square brackets with the use of a colour for the reference numbers. Eg [34]

Line 130: Change “ul” to “µL”

Changed

Reviewer 3 Report

As required for a literature review, the authors respected the specific rules  for this approach and the article is clear.
I just regret that in the overall discussion none of the works showing the impact of vector density on disease elimination possibilities was cited to enrich the discussion. For example https://dx.doi.org/10.3390/ijerph18010076 and others similar.

Best regards

Author Response

Reviewer three

Reference added

Round 2

Reviewer 2 Report

I have no new comments.  I just suggest authors to make a final revision for grammatical mistakes, typos, verify that all previous observations were attended. 

Author Response

Thank you, Reviewer for your valuable comments.

We have checked the paper through for grammar and typo errors
